

# Effects of self-isolation and quarantine on loot box spending and excessive gaming—results of a natural experiment

Lauren C. Hall[1], Aaron Drummond[1], James D. Sauer[2] and Christopher J. Ferguson[3]

[1] School of Psychology, Massey University, Palmerston North, Manawatu, New Zealand
[2] Psychology, School of Medicine, University of Tasmania, Hobart, Tasmania, Australia
[3] Department of Psychology, Stetson University, DeLand, FL, United States of America

## ABSTRACT

COVID-19 has prompted widespread self-isolation and citywide/countrywide lockdowns. The World Health Organisation (WHO) has encouraged increased digital social activities such as video game play to counteract social isolation during the pandemic. However, there is active debate about the potential for video game overuse, and some video games contain randomised purchases (loot boxes) that may psychologically approximate gambling. In this pre-registered study, we examined the effects of self-isolation and quarantine on excessive gaming and loot box spending. We recruited 1,144 (619 male, 499 female, 26 other) Australian, Aotearoa New Zealand, and US residents who self reported being quarantined or self-isolating ($n = 447$) or not ($n = 619$) during the COVID-19 pandemic to a cross-sectional natural experiment. We compared the associations between problem gambling symptomology, excessive gaming and loot box spending for isolated and non-isolated participants. Participants completed the Kessler-10 Psychological Distress Scale, Problem Gambling Severity Index, Internet Gaming Disorder Checklist, a measure of risky engagement with loot boxes, concern about contamination, and reported money spent on loot boxes in the past month, as well as whether they were quarantined or under self-isolation during the pandemic. Although, in our data, excessive gaming and loot box spending were not higher for isolated (self-isolated/ quarantined) compared to non-isolated gamers, the established association between problem gambling symptomology and loot box spending was stronger among isolated gamers than those not isolated. Concerns about being contaminated by germs was also significantly associated with greater excessive gaming and, to a lesser extent, loot box spending irrespective of isolation status. Gamers might be managing concerns about the pandemic with greater video game use, and more problem gamblers may be purchasing loot boxes during the pandemic. It is unclear whether these relationships may represent temporary coping mechanisms which abate when COVID-19 ends. Re-examination as the pandemic subsides may be required. More generally, the results suggest that social isolation during the pandemic may inflate the effect size of some media psychology and gaming effects. We urge caution not to generalise psychological findings from research conducted during the COVID-19 pandemic to be necessarily representative of the magnitude of relationships when not in a pandemic.

Corresponding author
Aaron Drummond,
A.drummond@massey.ac.nz

## INTRODUCTION

COVID-19 has prompted widespread non-pharmaceutical public health interventions including self-isolation and citywide lockdowns (*Lewnard & Lo, 2020*). Although these measures have many advantages, including a substantial flattening of the disease curve, they may have unintended maladaptive consequences. For instance, self-isolation and quarantine has substantially reduced access to safe and legal social activities. Social support is an essential aspect of mental health, and social isolation can increase psychological distress (*Teo, Choi & Valenstein, 2013*). Multiplayer video games offer one avenue for online social engagement. In fact, to encourage stay-at-home social activities, the World Health Organisation (WHO) recently partnered with 58 major video game companies to launch the #PlayApartTogether campaign, encouraging people to stay at home and play video games during the pandemic (*Takahasi, 2020*); despite having codified gaming disorder into their international classification of disease codes in 2018 (*World Health Organization, 2018*).

Scholarly debate persists about the potential for video games to foster maladaptive or excessive use (*Aarseth et al., 2017*; *Van Rooij et al., 2018*). In particular, there is significant disagreement about whether excessive gaming produces, or is a coping mechanism for, psychopathological symptoms such as depression and anxiety (*Van Rooij et al., 2018*). Irrespective of whether excessive gaming is a symptom or cause of psychopathology, there is clear theoretical rationale for examining the extent to which social isolation associated with pandemic control measures contributes to excessive gameplay. Moreover, people who are more concerned about being contaminated by the virus might also increase gameplay to cope with the anxiety or social isolation due to the pandemic. Here, we examined whether (a) socially isolated individuals exhibited higher rates of excessive gameplay than less isolated peers, and (b) whether contamination concerns (worry about, and avoidance behaviours of, disease and germs; *Burns et al., 1996*) were significantly associated with excessive gameplay.

A distinct but related issue is that the inclusion of certain monetisation mechanics within video games might contribute to excessive gameplay or spending (on game-related purchases). Some video games contain randomised purchasable rewards (loot boxes) that some scholars have observed may psychologically approximate conventional gambling activities (*Drummond & Sauer, 2018*; *Griffiths, 2018*). Some scholars have also observed that these mechanisms might meet the legal criteria in many jurisdictions, and could potentially be considered a form of bona fide gambling (*Drummond et al., 2020a*; *McCaffrey, 2019*). This may explain why studies have repeatedly found a small significant association between problem gambling symptomology and spending

on loot boxes (*Brooks & Clark, 2019*; *Drummond, Sauer & Hall, 2019*; *Drummond et al., 2020a*; *Drummond et al., 2020b*; *Li, Mills & Nower, 2019*; *Macey & Hamari, 2019*; *Zendle & Cairns, 2018*; *Zendle & Cairns, 2019*; *Zendle et al., 2020*), a result confirmed by meta-analysis (*Garea et al., 2020*). The decrease in alternative social activities, and the increased need for online social activities may have the potential to exacerbate this relationship. Spending on loot boxes therefore appears to be more common among video game players with problem gambling symptoms (*Garea et al., 2020*). The apparent increase in the number of people engaging with video games during the pandemic may result in increased exposure to, and hence opportunities for the purchasing of, loot boxes. This has prompted concerns about increased spending on loot boxes for certain user groups (*Department for Digital, Cultural, Media & Sport, 2020*; *Harris, 2020*). One study suggests that reducing exposure to loot boxes by removing them from a game appears to reduce spending only among those with higher problem gambling symptoms (*Zendle, 2019*). Thus it appears that problem gamblers who play games might be more likely to increase their spending on gambling-like features such as loot-boxes during the pandemic. This led us to predict that the social isolation of the pandemic might result in an increase in either overall spending on loot boxes, or, a specific increase in spending on loot boxes for problem gamblers. Thus, here we examined whether the established association between problem gambling symptomology was stronger for isolated gamers, compared to their non-isolated peers. As we thought that people who were more concerned about contamination would be more anxious and or more socially isolated during the pandemic than people who had lower contamination concern, we expected that people with higher contamination concern may have higher spending on loot boxes. Thus, we also examined whether contamination concerns were significantly associated with spending on loot boxes.

## Pre-registered hypotheses

In this pre-registered study (https://osf.io/f4cvj) we examined participants' excessive gaming scores, loot box spending, psychological distress, and whether they were self-isolating (limiting contact with others) or under quarantine (in mandatory self-isolation). We predicted that[1]:

1. Participants who report being in self-isolation or quarantine will have higher psychological distress (as measured by the K-10), higher spending on loot boxes, higher risky loot box use scores (as measured by the Risky Loot box Index; RLI), higher excessive gaming (as measured by the IGD scale), and higher contamination concern than participants not in self-isolation or quarantine.

2. There will be a significant positive correlation between contamination concern (as measured by the contamination subscale of the Revised Padua Inventory 10) and psychological distress as measured by the K-10.

3. There will be a significant positive correlation between contamination concern and spending on loot boxes, risky loot box use, and excessive gaming scores (as measured by the IGD scale). This effect will be stronger for participants currently in self-isolation or quarantine than for those not in self-isolation or quarantine.

[1]We intended to also assess the effect of being in citywide lockdowns, but the low number of participants reporting not being in lockdown ($n = 69$) rendered this analysis underpowered, likely invalidating the results. Interested readers can find these results in Supplemental Information.

4.  There will be a significant positive correlation between the amount of money participants report spending on purchasing loot boxes in the past month and their problem gambling symptoms as measured by the Problem Gambling Severity Index (PGSI).
5.  The relationship will be stronger for participants in self-isolation or quarantine than for those not in self-isolation or quarantine.
6.  Participants who are categorized as problem gamblers by the PGSI will report spending more money in the past month on loot boxes than participants who are categorized as moderate-risk gamblers, who in turn will spend more on loot boxes than low-risk gamblers, who in turn will spend more on loot boxes than non-gamblers.
7.  Being in self-isolation or quarantine will moderate the relationship between PGSI category and spending on loot boxes, such that problem gambler classification will show a larger difference in spending for participants currently in self-isolation or quarantine than participants not currently in self-isolation or quarantine.
8.  There will be a significant positive correlation between the amount of money participants report spending on purchasing loot boxes in the past month and psychological distress as measured by the Kessler Psychological Distress Scale (K-10).

## METHODS

### Participants

Using Prolific Academic, we employed a natural experimental design to compare those people who were isolated due to the pandemic to those people who had not been isolated by the pandemic. There was no theoretical reason to believe that the pandemic had selectively isolated gamers or problem gamblers at a higher rate than other people, so this created a natural-groups experiment to assess the effects of isolation due to the pandemic on the playing and spending habits of isolated (vs non-isolated) gamers. We therefore surveyed 1,200 participants from Australia, Aotearoa New Zealand and the US on their video game play and gambling. Excluding those who did not play video games ($n = 54$), and participants who failed attention checks ($n = 2$), our final sample was 1,144 participants (619 male, 499 female, 26 other). Table 1 shows the descriptive statistics of the sample by country.

The majority of participants were from the US ($n = 930$), with 173 participants coming from Australia, and 41 coming from Aotearoa New Zealand. Age ranged from 19 to 80 years old ($M = 31.4$ years, $SD = 10.5$ years). Participants gave informed consent to participate in the study by proceeding to the questionnaire after reading the information sheet which advised them to cease participation if they did not consent to participate in the study. Ethics approval was obtained for human data collection for this study from Massey University's Human Ethics Southern B Committee, Approval number SOB 19/11.

### Measures

Our study followed the same design as an earlier study protocol with the inclusion of some additional measures (*Drummond et al., 2020b*). We employed a variety of standardised psychiatric measurement tools to assess participants' wellbeing, problem gambling
**Table 1 Sample demographics split by country.** Participant numbers, mean age, and modal gender for participants in Aotearoa New Zealand, Australia and the United States.

| | Country | | |
|---|---|---|---|
| **Variable** | **Aotearoa New Zealand ($n = 41$)** | **Australia ($n = 173$)** | **United States ($n = 930$)** |
| **Mean age** | 34.4 ($SD = 10.2$) | 30.9 ($SD = 9.6$) | 31.3 ($SD = 10.7$) |
| **Modal gender** | Male ($n = 24$) | Male ($n = 109$) | Male ($n = 486$) |

symptomology, excessive gaming, risky loot box use and concern about contamination. These measures are detailed below.

### Psychological distress

The Kessler (K-10) Psychological Distress Scale assessed psychological distress of participants (*Slade, Grove & Burgess, 2011*) and was employed as previously described in *Drummond et al. (2020b)*. The K-10 includes 10 items assessing how often (0, none of the time–5, all of the time) over the past 30 days participants experienced various non-specific aspects of psychological distress (e.g., "During the last 30 days, about how often did you feel hopeless?").

### Spending on loot boxes in past month

Data on loot box spending was collected as previously described in *Drummond et al. (2020b)*. We asked participants to report approximately how much money (in their country's currency) they had spent on loot boxes in the past month. We converted all values into US dollars using the listed currency conversion rates of the day using Google's currency conversion. We converted all currencies into US dollars on April 21, 2020 using the following exchange rates: $USD = 0.63*AUD; $USD = 0.60*$NZD. In accordance with our pre-registration we excluded 9 participants (0.8% of total data) who spent more than 3.29 SDs (equivalent to a $Z$-score differing from the mean at $p < .001$; *Tabachnick, Fidell & Ullman, 2007*) greater than the mean ($6.08 USD, $SD = $42.50$ USD) as outliers from analyses of loot box spending. This resulted in the exclusion of any participant who spent more than $133.58 USD on loot boxes in the past month. Including these participants did not qualitatively alter the results.

### Risky loot box index

The Risky Loot Box Index (RLI), is a five item scale designed to examine risky engagement with loot box mechanics (*Brooks & Clark, 2019*). This scale was employed in the same manner as described in *Drummond et al. (2020b)*. The scale asks participants to rate their agreement on a standard 7 point likert scale with items assessing their cognitions associated with opening loot boxes (e.g., "Once I open a loot box, I often feel compelled to open another").

### Excessive gaming

Data were collected as previously described in *Drummond et al. (2020b)*. We adapted the Internet Gaming Disorder Checklist to assess excessive gaming symptoms (*Przybylski et al., 2017*). This checklist was based on the proposed diagnostic criteria for Internet Gaming

Disorder (IGD). Participants indicate how true (1, not at all true–4, very true) statements were about how gaming had interfered with aspects of their lives (e.g., "I have lost interest in other hobbies or entertainment in order to play games") or emotions (e.g., "I feel irritable, anxious or sad when I am unable to game").

### Contamination concern

We measured participants' contamination concern with the Revised Padua-Inventory 10 item contamination subscale (*Burns et al., 1996*). The Padua-Inventory 10 is a psychiatric scale assessing obsessive compulsive symptomology. The contamination subscale assesses the degree of disturbance a range of common activities which might result in contamination causes participants (0, not at all –4, very much). Example items include "I find it difficult to touch an object when I know it has been touched by strangers or by certain people" and "I feel my hands are dirty when I touch money."

### Problem gambling symptoms

Data on problem gambling symptoms was collected as previously described in *Drummond et al. (2020b)*. The Problem Gambling Severity Index (PGSI) assessed Problem Gambling Symptoms. This scale asks participants to indicate how frequently (0, never–3, almost always) during the past 12 months, they experienced problems caused by their gambling behaviors (e.g., "borrowed money or sold anything to get money to gamble"). The scale contains 9 items, with higher scores indicating greater symptoms of problem gambling. The PGSI has good validity in a non-clinical population (*Holtgraves, 2009*). The scale can also be used to categorize participants into discrete groups of varying risk for gambling problems. In accordance with *Currie, Hodgins & Casey (2013)* revised criteria for the PGSI which display better concurrent validity to the original scoring criteria, participants who score 0 on the scale are considered non-problem gamblers; low-risk gamblers score 1–4 on the PGSI; moderate-risk gamblers = 5–7 on the PGSI and problem gamblers score 8 or higher on the PGSI.

## Procedure

Participants completed the Kessler-10 Psychological Distress Scale (K-10; *Slade, Grove & Burgess, 2011*), Problem Gambling Severity Index (PGSI; *Currie, Hodgins & Casey, 2013*; *Holtgraves, 2009*), Internet Gaming Disorder Checklist (IGDC; *Przybylski et al., 2017*), a measure of risky engagement with loot boxes (Risky Loot Box Index; RLI; (*Brooks & Clark, 2019*), and reported money spent on loot boxes in the past month. Participants reported contamination concern on the Revised Padua Inventory-10 (*Burns et al., 1996*), and their isolation status: self-isolation ($n = 250$), quarantine ($n = 197$), or neither ($n = 697$). This created a natural-groups experiment to examine whether self-isolation or quarantine affected outcomes. We advertised our study on Prolific Academic as a study investigating video games and gambling. Data were collected between the 7th and the 9th of April 2020 (inclusive). At this time, Aotearoa New Zealand had been in stringent lock-downs (Level 4 under their COVID-19 Alert system) since the 25th of March 2020 (*Ardern, 2020*). Australia and the US had not enacted a federal lockdown response (though individual States had enacted different lockdown restrictions; *Morrison, 2020a*; *Schuchat, 2020*). For context,

Australia implemented a national boarder closure (with exemptions for returning citizens, permanent residents, and immediate family members) on 20th March 2020 (*Morrison, 2020b*), and non-US citizens travelling from China, Iran, The European Schengen Area, The United Kingdom and Ireland were banned from entering the US by Presidential Proclamations taking effect between 2nd February 2020 and 16th March 2020 (*Liu, 2020*; *NAFSA, 2020*).

We pre-registered comparisons between those who reported that they were self-isolating (eliminating contact with other people) or quarantined (under mandated self-isolation) and those who were not. We acknowledge that this grouping may overlook important differences in contexts between different forms of isolation (e.g., for participants anxiously awaiting test results), however two important points bear mentioning. First, the study was conducted early in the pandemic outbreaks in these countries, and terminology, especially among the general public, was relatively new, with exact restriction limitations still under development (*Cameron, 2020*) and some ambiguity in the Federal response in Australia and the US (*Morrison, 2020a*; *Schuchat, 2020*). We expected this would add noise to participants' self-reported isolation status, resulting in our decision to group all isolated participants together. Second, the self-isolation questions allowed participants to report whether they were in self-isolation or quarantine to their best understanding of these terms. This may result in some measurement error in reporting. However, we expected that variations in the accessibility of participants' usual social activities would be the primary driver of increased digital media exposure. Thus, we believe our approach offers a valid way to investigate the effects of increased perceived isolation. A more fine-grained study investigating whether effects are similar across different kinds of isolation using objective measures of such isolation would no-doubt be worthwhile. Nonetheless, our results provide early evidence that participants' perceived isolation status was associated with changes in some gaming-gambling relationships.

We also pre-registered comparisons between those who were under lockdown orders (specified to participants as "including mandated social distancing, business closures, curfews, stay home orders or other movement restrictions") in cities where movement and gathering restrictions were in place) and those who were not. However, very few participants were not in lockdown ($n = 69$, ~6% of our total data). Thus, estimates for non-lockdown participants are underpowered and, therefore, unreliable. Rather than report these unreliable estimates in the manuscript, we have included the results for these pre-registered comparisons between participants in lockdown and participants not in lockdown in the online supplementary materials with appropriate caveats. At the time of data collection, Aoteroa New Zealand had implemented a Nationwide lockdown which had been in effect for approximately 14 days. Federal lockdown responses had not been introduced in Australia, or the US, creating divergent state-based responses (*Morrison, 2020a*; *Schuchat, 2020*). We acknowledge that an examination of different stringencies of lockdown (e.g., Stage 2 versus Stage 4 lockdowns) would constitute a worthwhile follow-up study.

**Table 2** Associations between contamination concern and problematic gaming scales for all respondents (first column), participants not self-isolated or quarantined (second column), and self-isolated or quarantined (third column). $Z$-tests (column four) reported for the difference between not-isolated/quarantined vs self-isolated/quarantined. Correlations reported in Spearmans Rho.

| | Contamination Concern | | | |
| --- | --- | --- | --- | --- |
| | All respondents | Not self-isolated or quarantined | Self-isolated or quarantined | Z-test for difference |
| Loot Box Spend | .163[**] | .133[**] | .190[**] | $Z = 0.998$, $p = .319$ |
| RLI | .247[**] | .202[**] | .314[**] | $Z = 1.977$, $p = .048$[*] |
| IGD | .263[**] | .239[**] | .297[**] | $Z = 1.028$, $p = .426$ |

Notes.
[*] $p < .05$.
[**] $p < .001$.

## RESULTS

Participants played video games, on average, most days ($M = 2.31$, SD $= 0.83$). This was likely due to the fact that the study was advertised as a study on video game use and gambling, and therefore attracted participants who played video games. On average, participants reported spending $29.63 USD (SD $= $56.27$ USD; Range $= $0 - $500$ USD) on video games and $2.98 USD (SD $= $11.48$ USD; Range =$0 - $107.10$ USD) on loot boxes in the previous month. Although these averages appear small, it is worth noting that this was due to a large number of participants not purchasing video games ($n = 578$) or loot boxes ($n = 989$) in the past month, together with the wide ranges for these variables.

Contrary to predictions, psychological distress was actually higher among people not in self-isolation or quarantine ($M = 24.88$, SD $= 8.65$) than those in self-isolation or quarantine ($M = 23.53$, SD $= 8.68$), $t(1, 142) = 2.57$, $p = .010$, $d = 0.16$, though this was a negligible difference (*Ferguson, 2009*; *Sauer & Drummond, 2020*). We found no evidence that IGDC scores differed between participants in self-isolation/quarantine ($M = 7.94$, SD $= 5.59$) and participants not in self-isolation/quarantine ($M = 8.28$, SD $= 5.55$), $t(1, 142) = 0.97$, $p = .330$, $d = 0.16$. We also found no evidence of differences in spending on loot boxes between participants in self-isolation/quarantine (M $= $3.15$ USD, SD $= $11.38$ USD) and those not (M $= $2.86$ USD, SD $= $11.56$ USD), $t(1, 133) = 0.42$, $p = .678$, $d = 0.02$. There was no evidence of a difference in risky loot box use scores between participants who were isolating ($M = 13.53$, SD $= 8.16$) and those not ($M = 13.52$, SD $= 8.00$), $t(1, 142) = 0.02$, $p = .984$, $d = 0.01$. Similarly, contamination concern scores were not statistically different between those in self-isolation/quarantine ($M = 12.80$, SD $= 9.55$) to those not ($M = 13.77$, SD $= 9.72$), $t(1, 142) = 1.66$, $p = .098$, $d = 0.10$.

Collapsed across isolation status, contamination concern was moderately associated with distress, $r = .319$, $p < .001$, and weakly associated with loot box spending, IGDC, and RLI (see Table 2). Though weak, relationships between contamination concern and IGDC and RLI both exceeded criteria for the smallest effects likely to bear clinical relevance (*Ferguson, 2009*), suggesting contamination concern may be a clinically significant contributor to

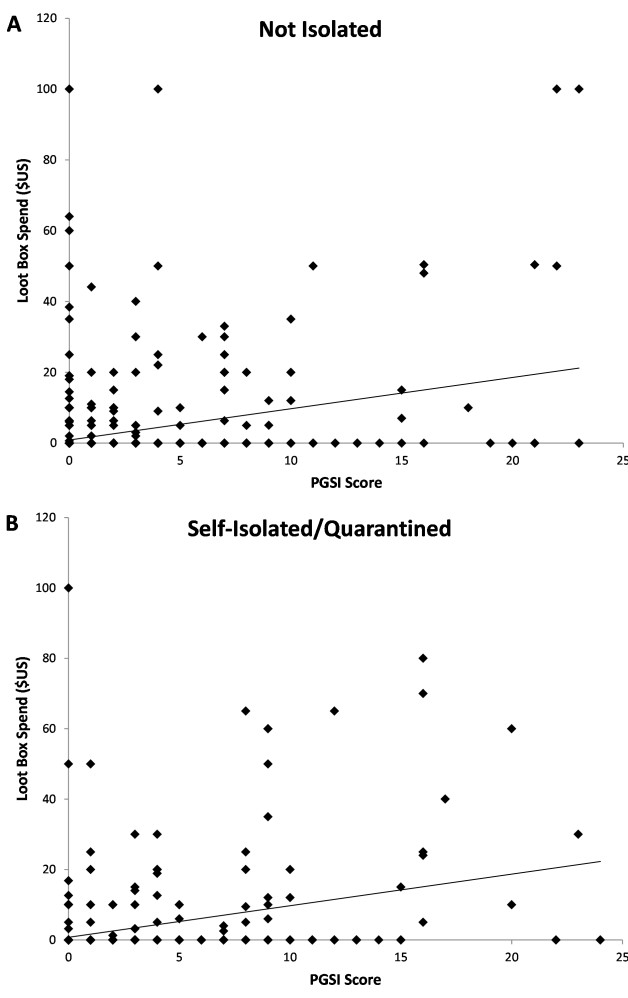

**Figure 1** **Association between Problem Gambling Symptomology (PGSI score) and Loot Box Spending in USD.** Data are presented separately for gamers who were not self-isolated/quarantined (A) and who were self-isolated/quarantined (B). Each data point represents a single participant's responses. The trend lines indicate the positive correlations between spending on Loot Boxes and PGSI scores, with a stronger association for participants who were self-isolated or quarantined than those who were not.

excessive engagement with both gaming and monetized in-game reward mechanisms. We compared correlation strengths using Fischer's $Z$-tests (*Weaver & Wuensch, 2013*). Relationships were generally similar for people in self-isolation or quarantine though the association between RLI and contamination concern was significantly stronger for participants who were self-isolated or quarantined than those who were not (Table 2).

Problem gambling symptomology was associated with higher loot box spending, $r_s = .279$, $p < .001$; with a stronger association for participants who were self-isolated or quarantined, $r_s = .366$, $p < .001$, than those who were not, $r_s = .209$, $p < .001$, $Z = 2.825$, $p = .005$. Figure 1 shows the association between loot box spending and PGSI scores, split by isolation status.

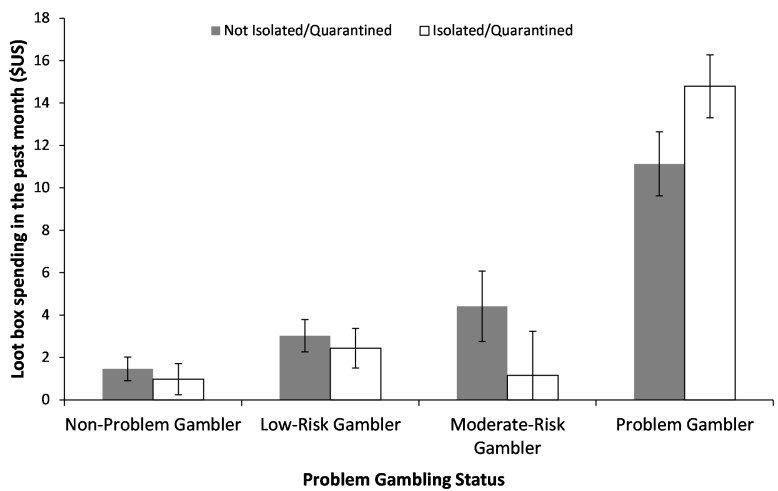

**Figure 2 Loot Box spending split by PGSI group and whether or not participants were Self-Isolated or Quarantined.** There was a significant main effect of PGSI group, $F(3, 1, 131) = 34.679$, $p < .001$. Planned comparisons showed that non-problem gamblers spent significantly less than low-risk gamblers, $t(955) = 1.987$, $p = .024$, $d = 0.170$, low-risk gamblers did not spend less than moderate-risk gamblers, $t(420) = 0.038$, $p = .485$, $d = 0.006$, but moderate-risk gamblers did spend less than problem-gamblers $t(185) = 5.986$, $p < .001$, $d = 0.548$. Error bars denote Standard Error.

When classified into risk groups, a 2-way ANOVA showed a main effect of problem gambler status, such that problem gamblers spent more than lower risk gamblers on loot boxes (Fig. 2). However, there was no evidence that self-isolation or quarantine altered this relationship. Finally, loot box spending was weakly associated with psychological distress, $r_s = .103$, $p < .001$.

## DISCUSSION

We investigated whether self-isolation or quarantine, or contamination concerns were associated with greater excessive gaming, loot box spending or psychological distress. Contrary to predictions, compared to those not self-isolated or quarantined, participants in self-isolation or quarantine showed negligiblly lower psychological distress, but no differences in loot box spending, loot box engagement, or excessive gaming. We found no evidence that self-isolation resulted in greater excessive gameplay or expenditure on loot boxes generally. It is unclear why lower psychological distress scores were observed for those in self-isolation or quarantine. Perhaps those in self-isolation or quarantine may perceive their health risks to be lower due to their isolation, and this offset any distress associated with isolation. Nonetheless, the effect size was negligible in magnitude, and unlikely to be of clinical importance (*Ferguson, 2009*).

However, higher contamination concern was associated with small-to-moderately higher excessive gaming and risky loot box engagement scores, and small increases in loot box spending. These effects were generally similar for participants who were self-isolating or in quarantine and those who were not, with the exception that the relationship between risky loot box use and contamination concern was stronger for people self-isolating or

in quarantine than those not. People may be engaging with video games to manage their contamination concerns about the COVID-19 pandemic. Of course, individuals may engage in a range of other activities (e.g., conventional gambling, online shopping, increased time streaming movies/ T.V. shows, etc.) as a means of managing contamination concern and social isolation due to the pandemic.

The established relationship between problem gambling symptomology and loot box spending (*Garea et al., 2020*) was stronger for those in self-isolation or quarantine than those not, demonstrating closer agreement between problem gambling symptom scores and loot box spending. Indeed the relationship between problem gambling symptomology and loot box spending was moderate for isolated gamers compared to small for non-isolated gamers.

However, when PGSI scores were used to categorise gamblers into discrete risk categories (e.g., low-risk, medium-risk, problem gamblers), average spending within categories was similar for self-isolated or quarantined participants and those not in isolation or quarantine. Since average spending did not increase for problem-gamblers in self-isolated or quarantined (vs. non self-isolated or non-quarantined), it appears that although the correlation between raw PGSI scores and loot box spending become more closely aligned for isolated gamers, how much they individually spend when devolved into risk categories does not substantially increase. *Orford et al. (2007)* suggest that gambling disorder is more appropriately viewed as a continuum rather than as discrete risk-groups, and the lack of differences between risk categories juxtaposed against a significant increase in the strength of the association between raw PGSI scores and spending may reflect a loss of variability in PGSI scores when this continuous measure is devolved into a categorical variable.

Whether these effects persist when self-isolation and quarantine periods end is unknown. These increases may plausibly represent temporary coping mechanisms which will abate when the threat of COVID-19 ends. Alternatively, given the prolonged pandemic activity worldwide, they may persist as the acquisition of new gambling-like behaviours. Re-examination as the pandemic subsides may be required to determine whether any effects are temporary or long lasting. If there are prolonged mental health effects of the pandemic, then this may plausibly result in similar long-term changes in the ways people interact with video games.

More generally, our results suggest that isolation due to the COVID-19 pandemic may act as a moderator for other relationships in pathological gaming research, inflating the size of some previously observed relationships (*Garea et al., 2020*). We therefore urge researchers to exercise caution in ensuring that the potential consequences of the pandemic and associated isolation are considered when interpreting the size of effects from data gathered during the pandemic in these areas of research. Further, we suggest that a moderating effect of COVID-19 may not be limited to these fields, and suggest that researchers in general carefully consider the potential impact of the pandemic on their data.

One limitation of the present study is that it is unknown whether participants were responding to the Padua Contamination Concern index as a state or trait based measure. It is thus unclear whether the associations between the contamination concern scale, loot box purchasing, excessive gaming and risky loot box use measures imply that changes

in concerns about contamination are causing changes in these outcomes, or if, more generally, people high in trait contamination concern tend to have higher scores on these measures. Our interpretation (and we admit that it is speculative) is that both may be true. Participants who are concerned about contamination may be particularly likely to engage in distracting behaviours (including spending behaviours) during a global pandemic. Future work should examine this possibility and its alternatives more deeply.

## CONCLUSION

This natural groups experiment examined eight hypotheses relating to the effects of the COVID-19 pandemic on excessive gaming, psychological distress, problem gambling and loot box spending. Our data suggest that contamination concern is associated with higher excessive gaming and risky loot box engagement, and to a lesser extent, loot box spending. Problem gambling symptomology is associated with greater loot box spending, and this effect is appreciably stronger among people who are currently self-isolating or in quarantine. Future work should examine whether the differences in the strength of the relationship between loot box spending and problem gambling symptomology for gamers who were isolated during the pandemic remains after the pandemic subsides.

### Funding
Supported was received from a Marsden Fund Council from Government funding (MAU1804), managed by Royal Society Te Apārangi awarded to Aaron Drummond, James D. Sauer, and Christopher J. Ferguson. The funders had no role in study design, data collection and analysis, decision to publish, or preparation of the manuscript.

### Grant Disclosures
The following grant information was disclosed by the authors:
Royal Society Te Apārangi Marsden Fund Council MAU 1804.

### Competing Interests
The authors declare there are no competing interests.

### Author Contributions
- Lauren C. Hall conceived and designed the experiments, performed the experiments, analyzed the data, prepared figures and/or tables, authored or reviewed drafts of the paper, and approved the final draft.
- Aaron Drummond, James D. Sauer and Christopher J. Ferguson conceived and designed the experiments, performed the experiments, analyzed the data, prepared figures and/or tables, authored or reviewed drafts of the paper, funding Acquisition, and approved the final draft.
## Human Ethics

The following information was supplied relating to ethical approvals (i.e., approving body and any reference numbers):

Massey University's Human Ethics Southern B Committee approved this research (approval number: SOB 19/11).

## Data Availability

Data is available at OSF:

Aaron Drummond, Lauren C. Hall, James D. Sauer, and Christopher J. Ferguson. 2021. "Loot Box Spending during a Pandemic." OSF. January 12. https://osf.io/bxf9m/.

## Supplemental Information

Supplemental information for this article can be found online at http://dx.doi.org/10.7717/peerj.10705#supplemental-information.

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
