# Peer review of "Effects of self-isolation and quarantine on loot box spending and excessive gaming—results of a natural experiment"

_PeerJ, doi:10.7717/peerj.10705_

## Round 0.1 · original submission · Major Revisions

Both reviewers found the study interesting and basically well-written but noted a number of places where the manuscript could be improved.

The reviewers have provided carefully-considered and concrete suggestions for improving the text, as well as for how the participants are described, how the hypotheses are analyzed, and how the results are presented. I would consider these suggestions carefully as you prepare a revised version.

Among their other comments, the reviewers suggest being clearer about the logic of the study rationale and hypotheses, and being clearer about which analyses correspond to which of the pre-registered hypotheses. I would echo Reviewer 2's suggestion of presenting the results in the same order as the pre-registered hypotheses are presented. Once the results based on pre-registered hypotheses (and pre-registered analysis plan) are presented, then if there are additional analyses you wish to conduct/present, those analyses should simply be identified as exploratory.

Both reviewers both also suggest additional theoretical considerations that could be considered or discussed in interpreting your results.

Finally, as noted by Reviewer 2, the (blank) consent form did not appear to be included in the supplemental material, per PeerJ requirements. Also as Reviewer 2 notes, the raw data on OSF should ideally be accompanied by a readme file with metadata and/or data dictionary.

Reviewer 1 ·

Basic reporting

Tables and graphics are good, but the writing and argumentative flow is a bit clunky in places, e.g., in the Abstract: “many have turned to digital social activities...’ (established?); the sudden jump to loot boxes seemed odd; the statement that one might find that isolation leads to excessive gaming seems to almost pre-empt the findings. Again, why was the main hypothesis all about the PGSI and loot boxes? The final two sentences in the Abstract also felt a bit unclear. P. 3 ‘In
fact,’ (the flow of writing could be better here).

p. 4 I struggled with the logic that set up the study. If lots of people start increasing their gambling why would you expect relationships to get stronger? Couldn’t you also get floor effects and less variability as the majority of gamers spend more on gaming?

p. 4 ‘suggesting that they were’ (try to avoid this construction)

Experimental design

See below. A few concerns about the design.

Validity of the findings

I had a few concerns about the paper.

First, the definition of isolation and quarantine is ambiguous. This has a specific meaning in disease management contexts. Isolation usually refers to situations where people who are sick are kept apart from other people, whereas quarantine usually refers to people who are kept apart from others because they may have been exposed, i.e., they may eventually show signs of the disease. The definitions here appear to mix up people who: (a) may be fearful and who are isolating because they may feel that they are vulnerable to COVID; (b) have just had a test and are told to stay home or have been put into self-isolation, but not necessarily in quarantine. These people might be getting police checks, but they are not really in quarantine; (c) People in actual quarantine, e.g., hotels, in hospital or other places. Lockdown could be Stage 4 or Stage 2. Big difference. For these reasons, I have concerns about the grouping that forms the basis of the analyses.
I also wondered- if you were in hotel quarantine – would you get to do as much gaming as when you were at home?

Second, the PGSI is described as classifying people as moderate risk based on scores of 5-7, but it’s actually 3-7? Low risk = scores of 1-2. Is the scoring in the study in accordance with the regular scoring?

Third, I found the hypotheses a bit odd. I was surprised that the main hypothesis wasn’t about increases in gaming or the relationship between gaming and loot box use. Problem gambling is most likely related to loot box use because people who enjoy gambling tend to gravitate towards gambling content in games. However, the main thrust of this paper is that the stress of COVID drives people towards greater gaming (as per the WHO recommendation- yes, ironic given the IGD classification, but sadly it doesn’t surprise me after their performance earlier this year!). In other words, wouldn’t one expect to see people’s gaming and loot box spending increase when they are really locked down and on mandatory stay-at-home orders? But it’s not so obvious that the cross over from gambling to gaming would occur.

Fourth, the study is all about moderation, but they haven’t really tested for moderation. It’s based on correlation difference tests. That’s OK, but it’s unusual. Maybe avoid referring to moderation and have a hypothesis that specifies that the simple first order effects will be different for the two groups.

Fifth, I didn’t feel that the paper provides enough detail about the demographic composition, gaming habits of the two groups; country etc..

Additional comments

Other general comments

p. 4 I struggled with the logic that set up the study. If lots of people start increasing their gambling why would you expect relationships to get stronger? Couldn’t you also get floor effects and less variability as the majority of gamers spend more on gaming?
p. 4 ‘suggesting that they were’ (try to avoid this construction)
p. 10 Another general point that needs to be made is that many people spent more on other things too: alcohol, take-away food. Netflix, etc. During the COVID period. It seems unlikely that a lack of access of gambling would increase loot box expenditure. There are many other reasons and people spend more on a range of things. People could have also increased their online gambling if they wanted to.

·

Basic reporting

I found the manuscript to be basically well written and well referenced.
As a comment on the introduction, I felt the authors tried to contextualize their study with regard to the controversy around the Internet Gaming Disorder diagnosis. E.g. line 58 “despite having codified gaming disorder (in the ICD-11)”, line 61 “significant disagreement”, and in particular, lines 64-67 that they are examining the effects of the pandemic on excessive gameplay “given the scientific debate about the potential for excessive video gameplay to have maladaptive consequences”. This needs to be handled quite carefully. Prima facie, it’s not clear to me why the effects of the pandemic on video gaming says anything about the IGD controversy (unless they are relying heavily on COVID-related contamination fears causing an increase in gaming, see comment in Validity). Second, the authors should pay careful attention to exact wording here, e.g. on line 65, for many academics, video gaming is only considered excessive if it has maladaptive consequences, so this line as worded is a tautology.
The study is pre-registered and the pre-reg hypotheses are presented on lines 94-123. I found it confusing that the results does not adhere to the same order of hypotheses. Either section could be re-organized but it would help the reader if they were aligned or at least cross-referenced.
I appreciate the dataset is open, but there is little reporting of descriptive stats other than in Figure 1. E.g. for interpreting the K-10 difference as ‘negligible’ (line 200 and 221) it would be great to see the means / SDs.
The authors present Fig 1 for the effects of self-isolation by PGSI group status. This analysis does not show a significant interaction. They do not show the corresponding figure for the PGSI continuous analysis that yields the significant interaction effect. One issue here is about the basic reporting, that is makes sense to graph the more interesting effect (or show both). There are related issues here that are perhaps more relevant to design and validity. First, in the pre-reg hypotheses, H3 is really the same hypothesis as H1, just tested with a categorical rather than continuous model. Likewise, H4 is the categorical version of H2. I'm not sure what to recommend here because these were pre-registered; but in both cases it is the same hypothesis tested in two ways. Second, the authors see an interaction (difference in correlation coefficients) in the continuous model (H2) but not the categorical model (H4). There is a speculative interpretation on line 243 about differences in the number of problem gamblers rather than how much they are spending, but the basic issue is that these 'discrepant' analyses are being conducted on the same data. It is more likely that this reflects either the thresholding of the PGSI (e.g. Currie et al thresholds vs the original Ferris & Wynne thresholds) or the relative statistical power of the two models. Inherently, a continuous model makes better use of the full data than any model that relies on binning scores together.
For PeerJ, the consent form should be uploaded with the Supp; the authors have confirmed this but I did not see it.
The raw data are available via an OSF link but there is no readme file on the variable annotations. With extensive reference to their pre-reg materials, I was able to find the geographical variable (see my comments on Design), but I would encourage the authors to prepare a document on the variable definitions.

Experimental design

This manuscript reports an online survey on Prolific of gaming behaviour during the early stage of the COVID pandemic. The focus on loot boxes (both expenditure and the Risky Loot Box index) and excessive video gaming provides important data for dissemination, and the design includes problem gambling (PGSI), distress (K-10) and contamination fears (Padua) as further covariates. The sample size is fairly large (n=1144) from US, New Zealand and Australia. A key finding is that the relationship between loot box spending and problem gambling is stronger in the group who describe themselves as self-isolating. This has implications for the public health discussion around the pandemic response, as well as wider implications for effect sizes in post-COVID gaming /gambling research.
I did have some concerns about the independent variable that forms the basis for the ‘natural experiment’ here. Participants were categorised as self-isolating or quarantining, versus not. A parallel model of lockdown (yes/no) was abandoned as too few participants endorsed being not in lockdown. Some basic points: 1) could the authors clearly state the date(s) on which the survey was run, perhaps in relation to important pandemic dates (e.g. declared state of emergency) in those jurisdictions, 2) can the authors add descriptives split by the 3 jurisdictions? 3) in the supp, can the authors state the number of participants who were in lockdown?
Currently, the analysis plan simply collapses by region. If for example NZ had a more restrictive ‘lockdown’ than the US, is it possible that their independent variable (self-isolation Yes/No) simply reflects a cross-jurisdictional effect? Second, the analysis plan considers the i.v. as ‘veridical’. Early in the pandemic there seemed to be public confusion about the terms ‘lockdown’ and ‘self-isolating’ (vs social distancing) (and again, the level of confusion may have varied by jurisdiction). The pre-reg materials do not indicate that definitions were provided to participants, which would be a limitation. Moreover, as 95% of participants were in lockdown, to what extent is ‘self-isolation’ within lockdown a subjective or objective status? (again I'm aware there are now clear jurisdictional definitions about these terms, but my point is whether the participants were thinking in these terms or not). This is quite important because subjective self-isolation (or conversely a subjective rejection of the local public health guidance) could be related to contamination fears in particular.

Other
Prolific survey: 54 participants did not play video games. How were the survey advertised on Prolific. As a prevalence rate, this seems high but it may reflect the general characteristics of Prolific participants. Please also add Age data to the Methods.
Line 170: why was the loot box outlier cut-off of 3.29 SD used? The authors state that this was a pre-registered threshold but it seems an unusual number.

Validity of the findings

The authors included measures of distress and contamination fears. The authors allude in several places to the debate about whether gaming causes ‘psychopathology’ or is used to copy with negative states, and their distress and contamination measures appear to be interpreted as speaking to this e.g. lines 42 and 44 in abstract. This interpretation seems to warrant some discussion as to whether these constructs are state or trait related. The K-10 is past month, so presumably state, but the Padua contamination scale could be viewed as a stable trait. The questionnaire did not appear to ask whether the items had changed at all in relation the pandemic to the pandemic. Similarly in discussion line 223 “isolation requirements have resulted in increased excessive gameplay or expenditure” – the verb “increased” here implies some longitudinal component or at least a retrospective judgment as to whether one’s activities have changed (i.e. under COVID). Their design does not speak to this.
Line 207 “small but important contributor” – how do Ferguson’s criteria establish that a small effect size is important?

Additional comments

no further comments

---

## Round 0.2 · accepted · Accept

Both reviewers feel, and I concur, that you have adequately responded to prior comments and that the paper is stronger as a result. (The reviewers have a few remaining minor suggestions which you may wish to consider.) The paper should make an interesting and topical addition to the literature.

Reviewer 1 ·

Basic reporting

Minor changes as indicated below

Experimental design

Issues addressed in revision

Validity of the findings

Issues addressed in revision

Additional comments

This is a thorough and well thought out revision. The rationale for the paper is much clearer in this revised version.

I found a couple of sentences to be a bit awkward:

"Thus it appears that spending by problem gamblers may be specifically impacted by the presence of loot boxes in video games while playing"

Comment: The structure of the sentence does not quite work and it's a key point. It needs to be something like:

Something like: "Problem gamblers who play games might be more likely to increase their spending on gambling-like features such as loot-boxes during the pandemic." ?


More generally, the results suggest that social isolation during the pandemic may inflate some pathological gaming effects

Comment: "pathological gaming effects" (odd expression). What is this effect? An increase in loot box expenditure in problem gamblers? An increase in the magnitude of the slope or strength of the relationship between PG and loot box spending? Need to make this a bit clearer.

·

Basic reporting

basic reporting is sound

Experimental design

I appreciate the authors' thorough response to the reviews. The comments on design and validity have been satisfactorily addressed.

Validity of the findings

See above. I have checked the new data dictionary on the OSF and I thank the authors for adding this document.

Additional comments

I have two remaining minor comments, both on the abstract
Abstract line 31 “…Australian, Aotearoa New Zealand, and US residents who were quarantined or self-isolating (n = 447) or not (n = 619)…” I would encourage the authors to be very clear here that participants were self-reporting as quarantining/self-isolating or not.
Also for the abstract line 44-45, I would question whether the sentence “These relationships may represent temporary coping mechanisms which abate when COVID-19 ends” is grounded in data from this study, and it seems at odds with the paragraph on line 327 of discussion. This sentence could be re-worded as "It is unclear whether these relationships..."